A transitional species of Daspletosaurus Russell, 1970 from the Judith River Formation of eastern Montana

Warshaw Elías A. 1 2 warshawelias@gmail.com
http://orcid.org/0000-0001-9188-4371 Fowler Denver W. 1
1 Badlands Dinosaur Museum , Dickinson, North Dakota , United States
2 Department of Earth Sciences, Montana State University , Bozeman, MT , United States
Farke Andrew
Electronic publication date: 2022 Nov 25
Publication date: 2022
Volume: 10
Electronic Location ID: e14461
Received 2022 Jul 27; Accepted 2022 Nov 3
Copyright: © 2022 Warshaw and Fowler
Copyright year: 2022
Copyright holder: Warshaw and Fowler
License: This is an open access article distributed under the terms of the Creative Commons Attribution License, which permits unrestricted use, distribution, reproduction and adaptation in any medium and for any purpose provided that it is properly attributed. For attribution, the original author(s), title, publication source (PeerJ) and either DOI or URL of the article must be cited.
License URL: https://creativecommons.org/licenses/by/4.0/

Keywords: Daspletosaurus, Anagenesis, Tyrannosaur, Paleontology

Funding: City of Dickinson, ND The US Bureau of Land Management TC Energy Conoco-Phillips TC Energy Employees Destiny and Craig Wolf Google Employees Funding for this project was provided by the City of Dickinson, ND; the US Bureau of Land Management, TC Energy, Conoco-Phillips, TC Energy employees, Destiny and Craig Wolf, Google employees. The funders had no role in study design, data collection and analysis, decision to publish, or preparation of the manuscript.

==============================
Here we describe a new derived tyrannosaurine, Daspletosaurus wilsoni sp. nov., from Judithian strata (~76.5 Ma) intermediate in age between either of the previously described species of this genus. D. wilsoni displays a unique combination of ancestral and derived characteristics, including a cornual process of the lacrimal reduced in height relative to D. torosus and more basal tyrannosaurines, and a prefrontal with a long axis oriented more rostrally than in D. horneri and more derived tyrannosaurines. The description of this taxon provides insight into evolutionary mode in Tyrannosaurinae, lending strength to previous hypotheses of anagenesis within Daspletosaurus and increasing the resolution with which the evolution of this lineage can be reconstructed. Cladistic phylogenetic methods, stratigraphy, and qualitative analysis of the morphology of relevant taxa supports an anagenetic model for the origin of morphological novelty in this genus, highlighting the predominance of anagenetic evolution among contemporary dinosaur lineages.

Introduction

Since their naming at the turn of the 20th century (Osborn, 1905), tyrannosaurids have captivated public and scientific imagination alike, and are as a result among the best-studied groups of Cretaceous theropods (Brusatte et al., 2010). Perhaps the most successful group of tyrannosaurids were the latest-Cretaceous tyrannosaurines, including among them a diverse array of forms from the slender-snouted alioramins (Lü et al., 2014) to robust and deep-jawed taxa like Teratophoneus (Carr et al., 2011) and the eponymous Tyrannosaurus rex (Carr & Williamson, 2010). However, much of the diversity of derived tyrannosaurines remains understudied or poorly understood (Paulina Carabajal et al., 2021), hampering understanding of paleobiogeographic and evolutionary trends (Loewen et al., 2013; Carr et al., 2017; Brusatte & Carr, 2016).

The tyrannosaurine Daspletosaurus has been known from Campanian fossil deposits of northern Laurasia for over half a century. However, published work on the phylogeny and paleobiology of this genus is relatively scarce beyond its initial description (Russell, 1970; Carr et al., 2017; Paulina Carabajal et al., 2021). Several enigmatic tyrannosaurine specimens initially referred to the type species or simply to Daspletosaurus sp. (including the recently named D. horneri) have been noted as representing novel species by previous workers for several decades (Carr, 1999; Currie, 2003; Carr et al., 2017; Horner, Varricchio & Goodwin, 1992; Paulina Carabajal et al., 2021), indicating a more speciose genus than has currently been described. Filling this gap is especially pertinent to understanding rates and patterns of speciation in the Campanian of Laurasia, both within tyrannosaurs and among dinosaurs as a whole, as both described species of Daspletosaurus have been hypothesized to represent an anagenetic lineage (Carr et al., 2017), including this genus among the many contemporary dinosaur lineages for which anagenesis has been suggested (Horner, Varricchio & Goodwin, 1992; Fowler & Fowler, 2020).

Here we describe Daspletosaurus wilsoni (sp. nov.). This addition to Campanian tyrannosaurid diversity has the potential to refine existing hypotheses regarding tyrannosaurid evolution in the Late Cretaceous, and lends strength to the hypothesis of anagenesis as a predominant mode of evolution in Daspletosaurus (Carr et al., 2017).

Tyrannosaurinae Matthew & Brown, 1922 (sensu Sereno, McAllister & Brusatte, 2005)

Daspletosaurus Russell, 1970

D. wilsoni sp. nov.

Etymology

wilsoni, Latinization of “Wilson,” after John Wilson, the discoverer of the holotype specimen.

Holotype

BDM 107, preserving a partial disarticulated skull and postcranium, including both premaxillae, a right maxilla, jugal, lacrimal, quadrate, quadratojugal, dentary, and splenial, and a left postorbital and squamosal. Also preserved are partial cervical, sacral, and caudal series, a rib, a chevron, and a first metatarsal. Cranial bones are very finely preserved, with intricate and detailed surface textures especially on the maxilla and postorbital, with teeth preserved in the maxilla, dentary, and one premaxilla. The sacral and caudal centra are preserved in a heavy and hard concretion and are not yet prepared. The holotype specimen is stored in the collections of the Badlands Dinosaur Museum (BDM) in Dickinson, North Dakota.

Geological Setting

BDM 107 was recovered from the site “Jack’s B2,” discovered in 2017 by John Wilson in exposures of the Judith River Formation near Glasgow (Valley County, MT, USA). This is significantly further east than classic ‘Judith’ localities (Fig. 1), and is sedimentologically atypical, representing distal floodplain and delta sediments deposited during the maximum Campanian regression of the Western Interior Seaway. Here, the Judith River Formation is up to ~48 m thick, with the “Jack’s B2” site occurring ~30 m below the contact with the overlying Bearpaw Shale.

Figure 1 Map of the area of discovery of BDM 107, holotype of D. wilsoni sp. nov.

Nearby towns (Hinsdale, Glasgow, Saco) and highways (US-2) are labeled. Dashed lines indicate county boundaries; “Jack’s B2” site indicated by star.

Precise stratigraphic placement of this easternmost Judith is currently unclear, although an age of ~76.5 Ma seems most likely, which would correlate in time with the lower to middle part of the Dinosaur Park Formation, Alberta (Eberth, Currie & Koppelhus, 2005; Fowler, 2017). A youngest age limit of 75.64 Ma (Ogg & Hinnov, 2012) is delineated by ammonites tentatively identified as Didymoceras stevensoni (J. Slattery, 2020, personal communication) collected by BDM from local outcrops of the overlying Bearpaw Shale (although these were not at the base of the Bearpaw, so older ammonite specimens may be encountered during future prospecting). At present, more precise stratigraphic position can be inferred from the timing of the maximum regression of the Western Interior Seaway during the Campanian (correlated with the R8 regression of Kauffman (1977) and Rogers et al. (2016)). In Alberta and Saskatchewan, the Foremost, Oldman, and Dinosaur Park formations represent early to late subcycles (respectively) of the R8 regression, and of these, the Foremost (~80.5–79.5 Ma) and lower Oldman (~79.5–79.0 Ma; and regional equivalents) are restricted to the west (Alberta and west central Montana), and did not extend as far east as Saskatchewan or our study area in eastern Montana (Eberth, Currie & Koppelhus, 2005). During late R8, the upper Oldman (~77.5–77.0 Ma) and Dinosaur Park (~76.9–76.0 Ma) Formations were deposited much further to the east, with the lowermost Dinosaur Park recording the R8 maximum regression at ~76.9–76.4 Ma (Eberth, Currie & Koppelhus, 2005; Fowler, 2017). This correlates well with the Judith River Formation of Montana, where Rogers et al. (2016) show the maximum regression of R8 occurring shortly before 76.2 Ma, based on radiometric dates acquired either side of the mid-Judith discontinuity. As such, it seems likely that the study section corresponds in age to the lower to middle part of the Dinosaur Park Formation (although not necessarily lithostratigraphically correlated). A radiometric analysis of a newly discovered volcanic ash is currently underway, and it is hoped that this will provide definitive stratigraphic placement.

Regardless of the precise age of BDM 107, it can be expected to lie intermediate stratigraphically between D. torosus (known from the upper Oldman Formation, ~77.0 Ma; Paulina Carabajal et al., 2021) and D. horneri (known from the Two Medicine Formation, ~75.0 Ma; Carr et al., 2017).

Diagnosis

D. wilsoni can be assigned to Daspletosaurus based on the following characteristics: extremely coarse subcutaneous surface of the maxilla with no elevated ridges or corresponding fossae (Carr et al., 2017; Voris et al., 2020); cornual process of the postorbital approaching the laterotemporal fenestra (Carr et al., 2017); dorsal postorbital process of the squamosal terminating caudal to the rostral margin of the laterotemporal fenestra (Carr et al., 2017; Voris et al., 2019); and extremely coarse symphyseal surface of the dentary (Voris et al., 2020).

D. wilsoni possesses a single autapomorphy: a rostrocaudally elongate and dorsoventrally narrow mylohyoid foramen of the splenial (this foramen is much deeper in other Daspletosaurus, Carr et al., 2017; see below), and can additionally be diagnosed by a unique combination of ancestral and derived Daspletosaurus characteristics. D. wilsoni and D. torosus share a pneumatic inflation of the lacrimal reaching the medial edge of the bone (this inflation does not reach the medial edge of the bone in the holotype of D. horneri, but this may represent an allometric, ontogenetic, or taphonomic bias; E. A. Warshaw, 2022, unpublished data; Carr et al., 2017), cornual process of the postorbital approaching the laterotemporal fenestra (this process terminates much more rostrally relative to the fenestra in D. horneri, contra Carr et al., 2017; see below), cornual process of the postorbital subdivided into two distinct processes (this subdivision is absent in D. horneri, E. A. Warshaw, 2022, personal observations; see Description); prefrontal oriented rostromedially (determined from the angle of the prefrontal articular surface on the lacrimal of the holotype of D. wilsoni, which does not preserve a prefrontal; the prefrontal of D. horneri is oriented mediolaterally), pneumatic excavation of the squamosal that does not undercut its rostromedial margin (entire margin undercut in D. horneri; Carr et al., 2017), and quadratojugal lacking a pneumatic foramen in its lateral surface (although the presence of this foramen is highly intraspecifically variable in both D. horneri and Tyrannosaurus, such that further discoveries of D. wilsoni individuals may reveal its presence in this taxon; Carr et al., 2017; Carr, 2020). D. horneri and D. wilsoni share, to the exclusion of D. torosus, a premaxillary tooth row oriented entirely mediolaterally, such that all but one premaxillary tooth is concealed in lateral view (rostromedial orientation in D. torosus and less derived tyrannosaurids), antorbital fossa of the maxilla terminating at the rostral limit of the external antorbital fenestra (this fossa extends ahead of this boundary onto the subcutaneous surface of the maxilla in D. torosus and less derived tyrannosaurids; Carr et al., 2017; E. A. Warshaw, 2022, unpublished data), rostrodorsal ala of the lacrimal inflated (uninflated in D. torosus and less derived tyrannosaurids), ventral ramus of the lacrimal longer than the rostral ramus (determined largely by the height of the postorbital bar in the reconstructed skull, given that the ventral ramus is largely unpreserved in the holotype of D. wilsoni; the rostral ramus of the lacrimal is longer than the ventral ramus in D. torosus; Carr et al., 2017), short cornual process of the lacrimal (tall in D. torosus, although this process is taller in D. wilsoni than D. horneri and may best be described as intermediate between the previously named species of this genus; Carr et al., 2017), and dorsal quadrate contact of the quadratojugal visible in lateral view (concealed in D. torosus and less derived tyrannosaurids).

Description

Given the wealth of detailed osteologies describing tyrannosaurine specimens (e.g., Carr (1999); Brochu (2003) and Hurum & Sabath (2003)), our description of the holotype of D. wilsoni places heavy emphasis on characteristics (or combinations of characteristics) unique to this specimen, as well as those that are otherwise taxonomically or phylogenetically informative within Tyrannosaurinae, so as to avoid the reiteration of plesiomorphic tyrannosaurine morphologies (or synapomorphies of Daspletosaurus) already described by previous authors (e.g., Carr et al. (2017); Voris et al. (2019) and Voris et al. (2020)).

Ontogenetic Stage of BDM 107

In order to facilitate comparison with other tyrannosaurine individuals of equivalent ontogenetic stages (and in doing so, to avoid the misattribution of a phylogenetic signal to ontogenetically derived characteristics), brief comment is warranted on the ontogenetic stage represented by BDM 107; two lines of evidence suggest that this specimen is of advanced ontogenetic age. Firstly, BDM 107 is among the largest known Daspletosaurus individuals (articulated skull length 105 cm; D. torosus holotype CMN 8506 skull length 104 cm, Voris et al., 2019; D. horneri holotype MOR 590 skull length 89.5 cm, Carr et al., 2017). Although Carr (2020) criticized the use of size as an indicator of ontogenetic status in Tyrannosaurus, this criticism was based on the absence of a correlation between size and maturity among adult individuals; all the largest specimens of this genus were unambiguously recovered as adult by Carr’s (2020) analysis (i.e., within the final stages of ontogenetic development), such that this feature remains ontogenetically informative in distinguishing adults from juveniles and subadults. Secondly, BDM 107 displays several morphologies known otherwise to characterize mature tyrannosaurines, including a deeply scalloped maxilla-nasal suture (Carr & Williamson, 2004; Carr, 2020), a maxillary fenestra positioned rostrally within the antorbital fossa (Carr, 2020), a cornual process of the lacrimal inflated and positioned dorsal to the ventral ramus (Carr, 1999; Currie, 2003; Carr, 2020), and a grossly exaggerated cornual process of the postorbital (Carr, 1999; Currie, 2003; Voris et al., 2019; Carr, 2020). The totality of this evidence supports an adult ontogenetic stage or later for BDM 107 (adult sensu Carr, 2020; ontogenetic Stage 4 sensu Carr, 1999); this hypothesis may be tested in future work through histological analysis and/or comparison with further discoveries of D. wilsoni individuals of different ontogenetic stages, both of which lie outside of the scope of the present study.

Premaxilla

The premaxillae of D. wilsoni are similar to those of D. horneri (Carr et al., 2017, Fig. 1; Fig. 2), Tarbosaurus (Hurum & Sabath, 2003, Fig. 3), and Tyrannosaurus (Brochu, 2003, Fig. 4) in that the alveolar row is oriented largely mediolaterally, such that the rostrum of the skull is broad and the labial surfaces of the premaxillary teeth face rostrally. In Tyrannosaurus and similarly derived tyrannosaurines (Tarbosaurus and D. horneri), the premaxillary teeth largely overlap each other in lateral view such that only the distalmost tooth is clearly visible; the same would be true of the holotype of D. wilsoni, were more than a single premaxillary tooth preserved within its socket. Conversely, the premaxillary tooth row of D. torosus and less derived tyrannosauroids is oriented rostromedially, such that multiple teeth are clearly visible in lateral view (Voris et al., 2019, Fig. 6).

Figure 2 Premaxillae of BDM 107.

Shown in lateral (A), medial (B), and rostral (C) views. Abbreviations are as follows: nf, neurovascular foramina; smp, symphysis. Scale is 10 cm.

Although previous authors have regarded a mediolaterally oriented premaxillary tooth row as a synapomorphy of Tyrannosauridae or more inclusive groups (e.g., Carr et al., 2017: character 15), this is in error; mature specimens of Gorgosaurus (UALVP 10, Voris et al., 2022, Fig. 1; AMNH 5458, Matthew & Brown, 1923, Fig. 2) and Qianzhousaurus (GM F10004, Foster et al., 2021, Fig. 2), have rostromedially oriented premaxillary tooth rows such that in specimens with preserved teeth, all premaxillary teeth are visible in lateral view (although all tyrannosaurids do have premaxillary tooth rows oriented more medially than basal tyrannosauroids; this is the phylogenetic signal recorded in character 15 of Carr et al. (2017)). Comparison with other tyrannosaurids is hampered by the absence of preserved premaxillae and/or published descriptions of this element for several species (e.g., Thanatotheristes, Voris et al., 2020; Dynamoterror, McDonald, Wolfe & Dooley, 2018; Nanuqsaurus, Fiorillo & Tykoski, 2014; Lythronax and Teratophoneus, Loewen et al., 2013, for which all published specimens lack premaxillae); however, D. wilsoni and more derived tyrannosaurines (D. horneri, Tarbosaurus, Tyrannosaurus) represent the greatest exaggeration of the medial inclination of the premaxillary tooth row among tyrannosaurids for which comparative material is available (although this condition, with only one clearly visible premaxillary tooth in lateral view, is present in at least one Gorgosaurus: TCMI 2001.89.1, Voris et al., 2022, Fig. 10). D. torosus is intermediate between the (presumably) ancestral rostromedial orientation and the mediolateral condition of later Daspletosaurus species; two to three premaxillary teeth are visible in lateral view in the holotype specimen, CMN 8506 (Carr & Williamson, 2004, Fig. 6; Voris et al., 2019, Fig. 6).

It should be noted that the orientation of the premaxillary tooth row is not necessarily equivalent to the orientation of the premaxillae themselves. In Tyrannosaurus AMNH 5027, for example, the premaxillae appear to be rostromedially oriented in dorsal view (Carr & Williamson, 2004, Fig. 7); however, the premaxillary alveoli are mediolaterally arranged when viewed ventrally (E. A. Warshaw, 2022, personal observations; Osborn, 1912, Fig. 5A; Molnar, 1991, Fig. 9A).

The taxonomic utility of this character is a hypothesis that will require further testing as individuals of D. wilsoni and other tyrannosaurids with preserved premaxillae are discovered; notably, two specimens previously referred to D. torosus display the derived condition (mediolateral orientation), sharing it with D. wilsoni and more derived tyrannosaurines: FMNH PR308 (Matthew & Brown, 1923, Fig. 5; Carr, 1999, Fig. 1) and TMP 2001.36.1 (Voris et al., 2019, Fig. 6). If these individuals were to represent D. torosus, the distinction between this species and D. wilsoni in the orientation of the premaxillary tooth row would be heavily undermined; however, both of these specimens have previously been noted as belonging to a novel taxon from the Dinosaur Park Formation (FMNH PR308, Currie, 2003; TMP 2001.36.1, Paulina Carabajal et al., 2021). Therefore, although relevant comparisons will be made with these specimens hereafter, they will be considered separately from D. torosus (and will be referred to below as the Dinosaur Park taxon). A precise taxonomic designation for these specimens is reserved for future work in accordance with comments by previous authors (Currie, 2003; Paulina Carabajal et al., 2021).

There is a small (~2 cm diameter) indentation in the nasal process of the right premaxilla of BDM 107; this is most likely pathological, as it is irregular in form and not present on the left premaxilla.

Maxilla

As in other Daspletosaurus, the subcutaneous surface of the maxilla in D. wilsoni is densely covered in anastomosing sulci extending from neurovascular foramina (Carr et al., 2017; Voris et al., 2020; Fig. 3). The degree of sculpturing of this surface in BDM 107 is similar to CMN 8506 (D. torosus), although in the former, there is no smooth region rostral to the external antorbital fenestra indicating a rostral continuation of the antorbital fossa as D. torosus and alioramins (Carr et al., 2017). As in Thanatotheristes and other Daspletosaurus species, the shallow excavations that characterize the maxillae of the most derived tyrannosaurines (Zhuchengtyrannus, Tyrannosaurus, Tarbosaurus; Hone et al., 2011; Voris et al., 2020) are absent from the holotype maxilla of D. wilsoni. Also absent are the textural ridges present on the maxillae of Zhuchengtyrannus (Hone et al., 2011), Tarbosaurus, Tyrannosaurus, and Thanatotheristes (Voris et al., 2020), but not any Daspletosaurus species.

Figure 3 Left maxilla of BDM 107.

Shown in lateral (A) and medial (B) views. Abbreviations are as follows: af, antorbital fossa; aof, antorbital fenestra; ma, maxillary antrum; mxf, maxillary fenestra; pmr, promaxillary recess; pmx, promaxillary fenestra; ps, palatal shelf; nf, neurovascular foramina; ns, neurovascular sulci. Scale is 10 cm.

The rostral end of the maxilla of BDM 107 is bowed subtly medially towards its contact with the premaxilla and nasal; this may be a structural consequence of the greater medial inclination of the premaxillary tooth row (see above), as a similar condition characterizes D. horneri (MOR 590, E. A. Warshaw, 2022, personal observations), Tarbosaurus (Hurum & Sabath, 2003, Fig. 15), and Tyrannosaurus (MOR 008, MOR 980, E. A. Warshaw, 2022, personal observations). Tyrannosaurids with more rostromedially inclined premaxillary tooth rows lack this bowing (e.g., D. torosus CMN 8506, J. T. Voris, 2022, personal communication).

The maxilla of BDM 107 is irregular relative to other species of Daspletosaurus in that it is proportionally elongate, being 64.1 cm in length and 24.8 cm in height (ratio of length to height = 2.6). This bone is 58.6 cm long rostrocaudally and 27.5 cm tall dorsoventrally in the holotype of D. horneri (ratio of length to height = 2.1; MOR 590, Carr et al., 2017). Given the broad range of variation in the proportions of this element in other tyrannosaurine species for which larger sample sizes are known (e.g., Tyrannosaurus; Carpenter, 1990; Paul, Persons & Van Raalte, 2022; E. A. Warshaw, 2022, personal observations), this characteristic was not included as an autapomorphy of D. wilsoni. Consistency in this trait across further discoveries of D. wilsoni individuals may require a reevaluation of the taxonomic utility of this character.

D. wilsoni possesses 15 maxillary alveoli, as in other species of Daspletosaurus (Carr et al., 2017). The 13th alveolus bears a swollen abscess in BDM 107, and the 15th maxillary tooth conceals a small replacement tooth within its root that is visible in medial (lingual) view. In general, the maxillary teeth are similar to those of other tyrannosaurid species in being labiolingually broad, although not to the degree present in more derived tyrannosaurines (e.g., Tyrannosaurus and Tarbosaurus), in which the labiolingual width of the maxillary teeth is subequal to their mesiodistal length (Carr et al., 2017). The first maxillary alveolus is not small and also bears an incrassate tooth (i.e., it does not bear a d-shaped crown similar to those present in the premaxillae, as in Gorgosaurus; Currie, 2003; Voris et al., 2022).

Jugal

The jugal of D. wilsoni is most similar to that of D. torosus among tyrannosaurines in that it has a mediolaterally thin ventral margin of the orbit (as opposed to a rounded margin as in Thanatotheristes, Lythronax, most Tarbosaurus, and some Tyrannosaurus; Voris et al., 2020; Voris et al., 2022; J. T. Voris, 2022, personal communication). A thin ventral margin of the orbit likely represents the ancestral tyrannosaurid condition, (as it is also present in Bistahieversor, Albertosaurus, Gorgosaurus, and D. horneri; J. T. Voris, 2022, personal communication) and does not bow medially along its rostrocaudal length (the jugals of D. horneri, Tyrannosaurus, and Tarbosaurus are angled rostromedially rostral to the orbit, such that the maxillae are medially inset from the orbitotemporal region; D. horneri MOR 590, E. A. Warshaw, 2022, personal observations; Tyrannosaurus AMNH 5027, Molnar, 1991, Fig. 9; Tarbosaurus GIN 107/1, Hurum & Sabath, 2003, Fig. 15; E. A. Warshaw, 2022, unpublished data).

As in D. torosus, the caudal portion of the lacrimal contact surface of the jugal is shallowly inclined (Fig. 4); this surface is very steep in D. horneri, as well as in Albertosaurus and Gorgosaurus (Carr et al., 2017). Although Carr et al. (2017) recovered this feature as unique to D. horneri among tyrannosaurines, it is also present in some Tyrannosaurus individuals (MOR 980, MOR 1125, AMNH 5027, E. A. Warshaw, 2022, personal observations).

Figure 4 Right jugal of BDM 107.

Shown in medial (A) and lateral (B) views. Abbreviations are as follows: cp, cornual process; po, pneumatic opening; pop, postorbital process. Scale is 10 cm.

Lacrimal

As in all tyrannosaurids except for D. horneri, Tarbosaurus, and Tyrannosaurus, the cornual process of the lacrimal in D. wilsoni is large and rises to a distinct apex along its dorsal margin (Carr et al., 2017; Fig. 5). This apex is situated directly dorsal to the lacrimal’s ventral ramus, as is characteristic of mature tyrannosaurines (Currie, 2003; Carr, 2020). The cornual process of the lacrimal is shorter in D. wilsoni (5.2 cm from the dorsal margin of the lacrimal antorbital recess to the apex of the cornual process in BDM 107) than D. torosus (6.9 cm, CMN 8506; Voris et al., 2019, Fig. 6), but similar to the Dinosaur Park taxon (5.1 cm, TMP 2001.36.1; Voris et al., 2019, Fig. 6) (these three specimens are each within 2 cm of each other in skull length, such that measurements of this process need not be corrected for differences in absolute specimen size; see above; Voris et al., 2019, Fig. 6). The lacrimal cornual process of the D. horneri holotype MOR 590 is shorter still (3.7 cm; Carr et al., 2017, Fig. 1), although it should be noted that this specimen is also ~15% shorter in skull length than any of the specimens previously mentioned (Carr et al., 2017; Voris et al., 2019; see above), such that the difference in this feature between D. horneri and other Daspletosaurus is relatively less pronounced than isolated measurements of this process would suggest (scaled isometrically to the same skull length as MOR 590, however, BDM 107 would still have a taller cornual process of the lacrimal, at 4.4 cm).

Figure 5 Left lacrimal of BDM 107.

Shown in lateral (A), medial (B), and dorsal (C) views. Abbreviations are as follows: cpa, cornual process apex; fr, frontal ramus; po, pneumatic opening; rda, rostrodorsal ala; rr, rostral ramus; vp, ventral process; vr, ventral ramus. Scale is 10 cm.

Carr et al. (2017) regarded an accessory cornual process of the lacrimal as a synapomorphy of Daspletosaurus. However, this process is indistinguishable from the caudally directed supraorbital process of the lacrimal upon which it is purported to sit; the supraorbital processes of the lacrimals of Tyrannosaurus (MOR 555, MOR 980, MOR 1125, AMNH 5027, E. A. Warshaw, 2022, personal observations), Tarbosaurus (ZPAL MgD-I/4, Hurum & Sabath, 2003, Fig. 6), and Teratophoneus (UMNH VP 16690, Loewen et al., 2013, Fig. 3) are all morphologically identical to those of Daspletosaurus, although they are scored by Carr et al. (2017) as lacking an accessory cornual process. In lieu of any quantitative demonstration of this process’s presence in Daspletosaurus, the taxonomic utility of this character is rejected here.

The lacrimal antorbital recess differs in morphology from D. torosus, but is similar to that of D. horneri, Tarbosaurus, and Tyrannosaurus in that the rostrodorsal ala joining the rostral and ventral rami of the lacrimal is inflated into a cylindrical bar that is elevated in relief relative to the rest of the recess (this ala is inflated in D. torosus, but to a lesser degree such that no discrete bar is formed between the rostral and ventral rami; Carr et al., 2017, Fig. S2F) (E. A. Warshaw, 2022, personal observations). This feature is also present in the Dinosaur Park taxon (TMP 2001.36.1, Voris et al., 2019, Fig. 6). Also distinguishing the lacrimal of D. wilsoni from D. torosus is a ventrally directed antorbital fossa in the latter. The lacrimal antorbital fossa is laterally directed in other tyrannosaurids, including D. wilsoni, the Dinosaur Park taxon (TMP 2001.36.1, Voris et al., 2019, Fig. 6), D. horneri (MOR 590 and MOR 1130, Carr et al., 2017, Fig. 3), Tyrannosaurus, Tarbosaurus, Albertosaurus, and Gorgosaurus (Carr & Williamson, 2004, Fig. 10).

Rostrally, the ventral process of the lacrimal rostral ramus is unique in D. wilsoni in having a rounded distal end; this process comes to a pronounced tip in most tyrannosaurids (Carr, Williamson & Schwimmer, 2005, Fig. 8; Loewen et al., 2013, Fig. 3), with the possible exception of D. horneri, in which the holotype specimen MOR 590 has a pointed ventral process and that of the paratype MOR 1130 is rounded (E. A. Warshaw, 2022, personal observations; Carr et al., 2017, Figs. 2C and 3). Given the eminent possibility of taphonomic alteration of this feature (i.e., “rounding down” of a pointed ventral process into a rounded one by abrasion prior to burial), exaggerated by the small size of the ventral process of the lacrimal, this feature is excluded from consideration either as an autapomorphy of D. wilsoni or as uniting this species with D. horneri.

Caudodorsally, the prefrontal articular surface of the lacrimal can be used to determine the orientation of the long axis of the prefrontal. In D. wilsoni and D. torosus, this element is oriented rostrocaudally (Carr & Williamson, 2004, Fig. 8). This condition is shared with the Dinosaur Park taxon (TMP 2001.36.1, Paulina Carabajal et al., 2021, Fig. 2D), and is also present in Gorgosaurus (UALVP 10, Voris et al., 2022), Teratophoneus (UMNH VP 16690, Loewen et al., 2013, Fig. 3), and Qianzhousaurus (GM F10004, Foster et al., 2021, Fig. 3). Conversely, the prefrontal is oriented rostromedially or mediolaterally in D. horneri (MOR 590, E. A. Warshaw, 2022, personal observations; Carr et al., 2017, Fig. 1), Tarbosaurus (ZPAL MgD-I/4, Hurum & Sabath, 2003, Fig. 1), and Tyrannosaurus (AMNH 5027, E. A. Warshaw, 2022, personal observations; Carr & Williamson, 2004, Fig. 8), as well as at least one specimen of Albertosaurus (TMP 1981.10.1, Carr & Williamson, 2004, Fig. 8).

Postorbital

The postorbital of D. wilsoni is most similar to that of D. torosus and the Dinosaur Park taxon in bearing a massive cornual process that approaches the rostral margin of the laterotemporal fenestra caudally (Fig. 6; Carr et al., 2017; D. torosus CMN 8506, Voris et al., 2019, Fig. 6; TMP 2001.36.1, Voris et al., 2019, Fig. 4). Carr et al. (2017) proposed a cornual process of the postorbital approaching the laterotemporal fenestra as a synapomorphy of Daspletosaurus; however, the cornual process of the postorbital does not approach the laterotemporal fenestra in the holotype of D. horneri (MOR 590, E. A. Warshaw, 2022, personal observations), and is instead broadly separated from it as in Tyrannosaurus (MOR 980, MOR 1125, MOR 555, E. A. Warshaw, 2022, personal observations) and Tarbosaurus (ZPAL MgD-I/4, Hurum & Sabath, 2003, Fig. 1).

Figure 6 Left postorbital of BDM 107.

Shown in lateral (A), dorsal (B), medial (C), caudal (D), and rostral (E) views. Abbreviations are as follows: cdt, caudodorsal tuberosity; dtf, dorsotemporal fossa; fc, frontal contact; lsc, laterosphenoid contact; sop, subocular process; sos, supraorbital shelf. Scale is 10 cm.

Also shared between D. wilsoni, D. torosus, and the Dinosaur Park taxon is the subdivision of the postorbital cornual process into two discrete processes: a supraorbital shelf protruding from the dorsal margin of the orbit and a caudodorsal tuberosity emerging more caudoventrally (Fig. 6; Voris et al., 2019, Fig. 4D), creating a sinusoidal relief when the postorbital is viewed rostrally or caudally. Both the supraorbital shelf and the caudodorsal tuberosity are situated upon a more ‘typical’ tyrannosaurine cornual process; that is, they lie lateral to a gross swelling of the postorbital similar to that present in other tyrannosaurines (e.g., Tyrannosaurus, MOR 1125, MOR 980, MOR 555, MOR 008, E. A. Warshaw, 2022, personal observations). The caudodorsal tuberosity overhangs its caudoventral base, creating a crease between this process and the underlying body of the postorbital; a similar condition is present in the postorbital cornual processes of Gorgosaurus, Teratophoneus, and Bistahieversor (Voris et al., 2022; J. T. Voris, 2022, personal communication), but not in D. horneri (MOR 590, Carr et al., 2017, Fig. 1), Tyrannosaurus (MOR 1125, MOR 980, MOR 555, MOR 008, E. A. Warshaw, 2022, personal observations), or Tarbosaurus (ZPAL MgD-I/4, Hurum & Sabath, 2003, Fig. 8). A similar crease forms between the body of the postorbital and the cornual process of Tyrannosaurus in specimens with an epipostorbital (sensu Carr, 2020; AMNH 5027, Molnar, 1991; Carr, 2020); however, no such element is present in the holotype of D. wilsoni (or any other Daspletosaurus specimens; E. A. Warshaw, 2022, personal observations).

The ventral ramus of the postorbital tapers ventrally to a point in D. wilsoni, as in other Daspletosaurus (Carr, 1999), including the Dinosaur Park taxon (Voris et al., 2019, Fig. 4), and in contrast to the enormous subocular process of the postorbital that projects rostrally in Tyrannosaurus (Carr, 2020), Tarbosaurus (Hurum & Sabath, 2003, Fig. 8), Gorgosaurus (Voris et al., 2022), Teratophoneus (Loewen et al., 2013, Fig. 3), and Albertosaurus (Currie, 2003). Although the subocular process is present in D. wilsoni (and other Daspletosaurus), it is small relative to those of other tyrannosaurids (Fig. 6).

Squamosal

The squamosal of D. wilsoni is indistinguishable from that of D. torosus in that the rostralmost extent of the postorbital contact surface terminates caudal to the rostral margin of the laterotemporal fenestra (also in D. horneri; Carr et al., 2017), the rostromedial margin of the pneumatic recess on the ventral surface is not undercut (Fig. 7), and the caudal process is pneumatized (as evidenced by pneumatic foramina in the process’s rostromedial surface; Carr et al., 2017). No characteristics or combinations of characteristics unique to D. wilsoni are observable on this element.

Figure 7 Left squamosal of BDM 107.

Shown in lateral (A), medial (B), and rostral (C) views. Abbreviations are as follows: cp, caudal process; ltf, laterotemporal fenestra; pcs, postorbital contact surface; po, pneumatic opening, qjp, quadratojugal process; rmm, rostromedial margin of pneumatic recess. Scale is 10 cm.

Quadratojugal

The quadratojugal is conservative morphologically across tyrannosaurids (Loewen et al., 2013). However, a single characteristic of the quadratojugal of D. wilsoni unites it with D. horneri and at least one specimen of the Dinosaur Park taxon (TMP 2001.36.1), and differs from the condition in D. torosus and less derived tyrannosaurids: a dorsal quadrate contact that is broadly visible in lateral view. In most tyrannosauroids, the dorsal quadrate contact of the quadratojugal is directed medially or rostromedially such that it is obscured by the body of the quadratojugal in lateral view. In D. wilsoni, D. horneri (MOR 590, Carr et al., 2017, Fig. 1), and TMP 2001.36.1 (Voris et al., 2019, Fig. 6), however, this process is directed caudomedially, exposing it laterally (Fig. 8).

Figure 8 Right quadratojugal of BDM 107.

Shown in lateral (A) and medial (B) views. Abbreviations are as follows: dqc, dorsal quadrate contact; jr, jugal ramus; sc, squamosal contact; vqc, ventral quadrate contact. Scale is 10 cm.

The dorsal quadrate contact is marginally visible laterally in the holotype of D. torosus, CMN 8506 (Voris et al., 2019, Fig. 6; J. T. Voris, 2022, personal communication), but not nearly to the extent observable in the aforementioned taxa. The condition in D. torosus may therefore represent individual variation on the caudomedial orientation of most tyrannosaurids, or a structural antecedent to the condition present in other species of Daspletosaurus.

The caudomedial orientation of the dorsal quadrate contact is reversed in the paratype specimen of D. horneri, in which this process is hidden in lateral view (MOR 1130, Carr et al., 2017, Fig. S2K). Given that this specimen is younger stratigraphically than the holotype (MOR 590; Carr et al., 2017), this reversal may represent a phylogenetic signal (although it may instead represent intraspecific variation). Tarbosaurus and Tyrannosaurus share this feature with MOR 1130 (E. A. Warshaw, 2022, personal observations).

Quadrate

No discrete morphological characters distinguish the quadrate of D. wilsoni from those of its closest relatives. As in other derived tyrannosaurines, the quadrate is massive, with a shallow fossa on its medial surface and a pronounced pneumatic foramen (and surrounding fossa) at the rostral confluence of the mandibular condyles and the orbital process (Fig. 9; Carr et al., 2017). The paraquadrate foramen, bounded medially by the quadrate and laterally by the quadratojugal, is small and teardrop-shaped; only its lateral margin is made up by the quadratojugal, as the quadrate forms the dorsal and ventral borders of the foramen.

Figure 9 Right quadrate of BDM 107.

Shown in medial (A) and lateral (B) views. Abbreviations are as follows: op, orbital process; mc, mandibular condyles; po, pneumatic opening; pqf, paraquadrate foramen. Scale is 10 cm.

Although no palatal elements are known, the medial deflection of the quadrate’s pterygoid wing allows an approximation of the position of the pterygoids relative to the facial skeleton, and suggests a broad orbitotemporal region, as in other tyrannosaurines.

Dentary

The dentary of D. wilsoni is deep, with a relatively straight ventral margin and a dorsal (alveolar) margin that trends caudodorsally, increasing the depth of the mandible caudally (Fig. 10). As in other Daspletosaurus, the texturing of the dentary symphysis is more exaggerated in D. wilsoni than non-Daspletosaurus tyrannosaurines (e.g., Tyrannosaurus, Thanatotheristes; Voris et al., 2020), and is composed of several interlocking (presumably, as only the left dentary is known) ridges and cusps. There are 17 dental alveoli, as in D. horneri (Carr et al., 2017), and a sharp, narrow Meckelian groove with a rugose knob caudoventral to its rostral end. This knob is present in both other species of Daspletosaurus, as well as Tyrannosaurus, Tarbosaurus, and Zhuchengtyrannus magnus, but not Thanatotheristes or more basal tyrannosaurids (Carr et al., 2017; Voris et al., 2020).

Figure 10 Right dentary of BDM 107.

Shown in lateral (A) and medial (B) views. Abbreviations are as follows: dc, dentary chin; dg, dentary groove; mcf, Meckelian foramen; mg, Meckelian groove; nf, neurovascular foramina; ns, neurovascular sulci; pt, pathology; rk, rugose knob. Scale is 10 cm.

The lateral surface of the dentary of BDM 107 bears two intersecting grooves caudoventral to the caudal termination of the alveolar margin (Fig. 10); the edges of these grooves are beveled and are likely pathological. They may represent bite marks, as have been described on the craniofacial bones of other tyrannosaurids (Voris et al., 2020; Brown, Currie & Therrien, 2022).

Splenial

The splenial of BDM 107 is typical of Daspletosaurus except in the size and form of the mylohyoid foramen (Fig. 11), an autapomorphy of this taxon. In most derived tyrannosaurines, including D. torosus and D. horneri, this foramen is extremely large, roughly the same dorsoventral depth as the rostral process of the splenial (Carr et al., 2017). In D. wilsoni, however, the foramen is dorsoventrally shallow, and rostrocaudally elongate, such that it is ellipsoid in form and roughly half the dorsoventral depth of the splenial’s rostral process. This is most similar to the condition in alioramins (Brusatte, Carr & Norell, 2012) and Appalachiosaurus (Carr, Williamson & Schwimmer, 2005).

Figure 11 Right splenial of BDM 107.

Shown in medial (A) and lateral (B) views. Abbreviations are as follows: dcs, dentary contact surface; mhf, mylohyoid foramen. Scale is 10 cm.

Cervical vertebrae

Four cervical vertebrae are preserved in BDM 107 from the cranial-middle portion of the series. No atlas or axis were found. As in all tyrannosaurids, the spinous processes of the cervical vertebrae are subequal in dorsoventral height to their corresponding centra. Both the spinous processes and the centra are craniocaudally short, similar to and most exaggerated in the cervical vertebrae of Tyrannosaurus (see Brochu, 2003, and figures therein). As in Tyrannosaurus (and other large tyrannosaurids), the cranial and caudal faces of the cervical centra in BDM 107 are dorsoventrally displaced from one another in order to create the characteristic ‘S-curve’ of the neck, and the cranial cervical centra are extremely foreshortened craniocaudally (i.e., much taller than long). This indicates a robustly built cranial portion of the neck, presumably in order to support the weight of the head.

Sacral vertebrae

The spinous processes of two sacral vertebrae are preserved. Both are sub-rectangular in form and bear rugose knobs near their apices, presumably the ossified bases of sacral ligaments.

Methods

The holotype specimen was collected under permit MTM 108829-e6 issued to DF by The US Bureau of Land Management.

The electronic version of this article in Portable Document Format (PDF) will represent a published work according to the International Commission on Zoological Nomenclature (ICZN), and hence the new names contained in the electronic version are effectively published under that Code from the electronic edition alone. This published work and the nomenclatural acts it contains have been registered in ZooBank, the online registration system for the ICZN. The ZooBank LSIDs (Life Science Identifiers) can be resolved and the associated information viewed through any standard web browser by appending the LSID to the prefix http://zoobank.org/. The LSID for this publication is: urn:lsid:zoobank.org:pub:F7EE2619-89FC-4D72-93DA-EFE6BD549A77. The online version of this work is archived and available from the following digital repositories: PeerJ, PubMed Central SCIE and CLOCKSS.

A cladistic phylogenetic analysis was conducted using the character matrix of Carr et al. (2017) (with modifications from Voris et al. (2020)), with additional modifications based on personal observation of specimens made by the lead author, including the addition to the character matrix of several proposed autapomorphies of D. horneri noted by Carr et al. (2017) to occur more broadly across Tyrannosauridae (see Supplemental Information for a comprehensive list of modifications). The analysis was run in TnT v1.5 (Goloboff, Farris & Nixon, 2008) using a “New Technology” search with settings identical to those of Voris et al. (2020) (ratchet, tree drift, tree fusing, and sectorial search set to default, and set to recover minimum length 10 times). Support for recovered clades was tested using bootstrapping with 1,000 replicates under a traditional search.

Results

The cladistic analysis produced 12 Most Parsimonious Trees (MPTs; best score: 853). Within the strict consensus of these trees, the least inclusive clade containing Dynamoterror and Tyrannosaurus (i.e., all of Tyrannosaurinae more derived than Alioramus) was recovered as a large polytomy, with a sister relationship retained between Tyrannosaurus and Tarbosaurus, and Dynamoterror, Lythronax, and Teratophoneus recovered in a trichotomy (see Supplemental Information: Fig. S1).

Given the fragmentary nature of their respective holotypes (scored for <15% of characters), Nanuqsaurus hoglundi and Thanatotheristes were removed from the dataset (inclusion of either taxon collapsed the tree as above), and an additional analysis was conducted with the same settings. This analysis produced two MPTs (best score: 846), and recovered D. wilsoni as sister to a clade formed by D. horneri and more derived tyrannosaurines (Zhuchengtyrannus, Tarbosaurus, Tyrannosaurus). Alioramins were recovered within a polytomy, as were Dynamoterror, Teratophoneus, and Lythronax; all other topological relationships were as in Voris et al. (2020) (Fig. 12).

Figure 12 Results of the cladistic analysis.

Grey nodes denote Daspletosaurus, star denotes D. wilsoni, and numbers by each node are bootstrap support. Skull reconstruction represents the holotype of D. wilsoni, BDM 107 (known material in white).

Bootstrapping of this result showed weak support (<70) for all clades within Tyrannosaurinae except for alioramini (90), derived tyrannosaurines (Daspletosaurus + (Zhuchengtyrannus (Tyrannosaurus + Tarbosaurus))) (82), tyrannosaurines more derived than Daspletosaurus (85), and Tyrannosaurus + Tarbosaurus (85). Recovered support was particularly weak (≤9) for the interrelationships of Daspletosaurus (Fig. 12).

A single autapomorphy of D. wilsoni was recovered by the cladistic analysis: mylohyoid foramen of the splenial elongate and rostrocaudally ovoid (this foramen is much deeper in other Daspletosaurus species; see above).

The D. wilsoni + more derived tyrannosaurines clade was recovered with the following three synapomorphies: dorsoventrally tall orbit; mediolaterally oriented tooth row of the premaxilla; and short cornual process of the lacrimal. A further four synapomorphies united D. horneri and more derived tyrannosaurines to the exclusion of D. wilsoni: rostrolaterally directed orbits (resulting from the rostromedial bowing of the jugal); cornual process of the postorbital swollen and terminating far rostral to the laterotemporal fenestra; first interdental plate of the maxilla narrow, and second plate truncated (both plates are subsequently expanded in tyrannosaurines more derived than D. horneri); and mediolaterally oriented prefrontal. Additional autapomorphies of relevant taxa and synapomorphies of relevant clades are available in Supplemental Information.

Discussion

Several aspects of the results presented here contrast with (or supplement) those of previous analyses, and therefore deserve mention. Noticeably, the results of the cladistic analysis place Tyrannosaurus—line tyrannosaurines (Zhuchengtyrannus, Tarbosaurus, and Tyrannosaurus) as successive sister taxa to Daspletosaurus (contra Carr et al., 2017, and Voris et al., 2020, both of which recovered these as sister lineages), and recovers a paraphyletic Daspletosaurus; these aspects of the results are the topic of a study by the lead author currently in review, and will not be discussed here (although it should be noted that similar results were recovered by Horner, Varricchio & Goodwin (1992), and the Bayesian analysis of Brusatte & Carr (2016), Loewen et al. (2013) also recovered a paraphyletic Daspletosaurus. Should this paraphyly be upheld by future studies, D. wilsoni and D. horneri may be assigned new genera in order to preserve monophyly; D. wilsoni is assigned to Daspletosaurus here to avoid the creation of a polyphyletic Daspletosaurus and for ease of discussion and comparison with its closest relatives). Instead, only the interrelationships and evolutionary history of Daspletosaurus are considered below.

Though not included in the cladistic analysis, the Dinosaur Park taxon agrees with D. wilsoni in several characters which differ in both of these taxa from the condition in D. torosus (see Description), including the orientation of the premaxillary tooth row, the height of the cornual process of the lacrimal, the inflation of the rostrodorsal ala of the lacrimal, and lateral exposure of the dorsal quadrate contact of the quadratojugal. All of these characters are also shared with D. horneri, although D. wilsoni and the Dinosaur Park taxon also share (to the exclusion of D. horneri) a cornual process of the postorbital that approaches the laterotemporal fenestra and is subdivided into a caudodorsal tuberosity and a supraorbital shelf, and a prefrontal that is oriented rostrocaudally rather than rostromedially or mediolaterally (both of these characters are also present in D. torosus). Similarity in all of these features suggests a close affinity between D. wilsoni and the Dinosaur Park taxon, although this could reflect either taxonomic synonymity or a genuine sister relationship; this designation is reserved for future studies centered on the Dinosaur Park taxon (noted as forthcoming by Currie (2003) and Paulina Carabajal et al., 2021), which has yet to receive a formal description and may reveal autapomorphies (or synapomorphies with D. wilsoni) not considered here.

Should the Dinosaur Park taxon be demonstrated to represent a distinct species from D. wilsoni, it would potentially represent the first known instance of contemporaneity between more than one species of Daspletosaurus (Carr et al., 2017), given that the D. wilsoni holotype was preserved in strata likely corresponding in time to the deposition of the Dinosaur Park Formation (at least in part; see Geologic Context). However, this possibility rests both upon the taxonomic distinctiveness of the Dinosaur Park taxon and the absence of fine-scale stratigraphic separation between this species and D. wilsoni, both of which require additional study to confirm or deny (e.g., a formal description of the anatomy of the Dinosaur Park taxon and precise stratigraphic placement of individuals of this taxon and D. wilsoni). Discussion below will therefore exclude this possibility from consideration, although resulting hypotheses will be subject to revision should this exclusion prove to be erroneous.

Among described Daspletosaurus species, D. wilsoni fulfills the predictions made by Carr’s et al. (2017) hypothesis of anagenesis between D. torosus and D. horneri. Namely, D. wilsoni is stratigraphically, phylogenetically, and morphologically intermediate between these taxa (see Geologic Context, Results, and Diagnosis, respectively), and occurs within the same general geographic range (all three species of Daspletosaurus are found within Montana or Alberta; Carr et al., 2017). These points correspond to the criteria proposed by Carr et al. (2017) (and later Zietlow, 2020) for defensible hypotheses of anagenesis: (1) lack of stratigraphic overlap (but see above), (2) close phylogenetic relationships, (3) intermediate morphologies, and (4) similar geographic ranges. It should be noted that while the fulfillment of these criteria establishes anagenesis as a defensible hypothesis, it does not preclude cladogenesis in Daspletosaurus as the driving factor of the evolution of this genus (with successively more derived clades, e.g., D. wilsoni and more derived tyrannosaurines, representing cladogenetic events rather than portions of an anagenetic sequence).

However, several alternative lines of evidence are consistent with anagenesis and inconsistent with cladogenesis, and therefore strengthen the hypothesis of anagenesis as a predominant evolutionary mode in Daspletosaurus. Firstly, as noted by Wagner, Erwin & Anstey (1995), cladogenesis via punctuated equilibrium (with species diverging from an ancestral taxon in morphological stasis; Eldredge & Gould, 1972) can be identified by the presence of polytomies in a recovered cladogram, since descendant species of an ancestor in stasis will not form subclades. No polytomy was recovered within the clade formed by Daspletosaurus and more derived tyrannosaurines (Fig. 12); the origination of sampled Daspletosaurus species from a common ancestor in stasis can therefore be rejected based on the topology of the recovered cladogram alone. Secondly, D. wilsoni almost entirely lacks autapomorphies, displaying only a single feature not also present in either D. torosus or D. horneri (a rostrocaudally elongate mylohyoid foramen of the splenial, recovered by the cladistic analysis; see Results). Wagner, Erwin & Anstey (1995) and Szalay (1977) noted that ancestors should lack apomorphies relative to descendants, such that a paucity of autapomorphies suggests that an ancestral taxon has been sampled. Similarly, Wilson, Ryan & Evans (2020) noted the absence of autapomorphies in several centrosaurine taxa hypothesized therein to represent an anagenetically evolving lineage, with stratigraphically successive taxa being defined by combinations of plesiomorphic and apomorphic characters rather than species-level autapomorphies (forming “metaspecies;” Horner, Varricchio & Goodwin, 1992; Wilson, Ryan & Evans, 2020). Indeed, a hypothesis of cladogenesis at the root of the clade formed by D. wilsoni and more derived tyrannosaurines would rest entirely upon the elongate mylohyoid foramen of this species as evidence of divergence from other Daspletosaurus; in the absence of additional characters supporting this hypothesis, the sole autapomorphy of D. wilsoni may alternatively represent individual variation, or a character evolved within this species and lost before the appearance of D. horneri (similar to the lateral exposure of the dorsal quadrate contact of the quadratojugal, present in D. wilsoni and the holotype of D. horneri, but not in the stratigraphically sequential paratype specimen or more derived tyrannosaurines; see Description). The morphological evidence for a cladogenetic origin of D. wilsoni is therefore weak; the blend of ancestral and derived characteristics in this species and the near total absence of autapomorphies is more consistent with anagenesis between stratigraphically antecedent (D. torosus) and subsequent (D. horneri) taxa.

In light of this evidence, we propose that the three species of Daspletosaurus represent an anagenetically evolving lineage (Fig. 13); as noted above, this hypothesis will be subject to revision following further study into the phylogenetic affinities of species within the genus, additional discoveries of Daspletosaurus individuals from stratigraphically intermediate horizons (which under an anagenetic model, should be intermediate in morphology between species), and characterization of the range of individual variation present in relevant characters proposed here to represent species-level autapomorphies or morphological transitions between taxa.

Figure 13 Time-calibrated sequence of Daspletosaurus chronospecies.

Ages (left) are in Ma and are based on Carr et al. (2017) and Fowler (2017) for D. torosus and D. horneri. Representative skulls are, from top to bottom: D. horneri, MOR 590; D. wilsoni, BDM 107 (known material in white); D. torosus, CMN 8506. Stars represents the temporal position of adjacent specimens. Accompanying characters represent synapomorphies of progressively more exclusive clades represented by each taxon (e.g., D. wilsoni + more derived tyrannosaurines, D. horneri + more derived tyrannosaurines). No clear demarcations are drawn between taxa along the depicted lineage, given the relative paucity of specimens and the subjectivity intrinsic to species delineations of anagenetic lineages; ages of taxa are therefore imprecise. Scale is 10 cm.

Should branching events (i.e., cladogenesis) within Daspletosaurus be demonstrated by future studies or discoveries (e.g., if the Dinosaur Park taxon is demonstrated to be both distinct from and contemporaneous with D. wilsoni), this would not necessarily exclude anagenesis from playing a role in the generation of morphological novelty within the genus. Wagner, Erwin & Anstey (1995) noted the presence of anagenetic change between branching events in plesiomorphic lineages (=ancestral lineages; the lineage from which cladogenetically derived taxa branch) not in morphological stasis, which led these authors to designate this pattern of speciation as “bifurcation,” reserving “cladogenesis” for branching from morphologically static ancestral taxa. Although we do not adopt their terminology, we agree that anagenesis can operate in concert with cladogenesis in order to produce observed patterns of macroevolutionary change. In the case of Daspletosaurus, while autapomorphies of individual species may represent the results of cladogenesis, the synapomorphies of progressively more exclusive clades within the genus (e.g., coarse symphyseal texture of the dentary in Daspletosaurus, inflated rostrodorsal ala of the lacrimal in D. wilsoni + D. horneri, etc.) would remain anagenetically derived under a typically cladogenetic model. Anagenesis therefore enjoys a predominant role in the evolution of derived morphologies within derived tyrannosaurines, regardless of the presence of branching events within Daspletosaurus (in contrast to morphologically static genera, in which morphological change is concentrated at the base of cladogenetic events; Eldredge & Gould, 1972).

The low bootstrap support recovered for the results of the cladistic analysis may also be readily explained in the context of anagenesis. As noted by Soltis & Soltis (2003), low bootstrap scores may be recovered for otherwise well-supported clades (e.g., clades recovered within all MPTs, as in all of the interrelationships of Daspletosaurus recovered here) if they are supported by few characters, given that the chance of supporting characters being included in a bootstrap resample is lower with fewer characters. This is a common occurrence among recently diverged clades which have not had much time to accrue synapomorphies (Soltis & Soltis, 2003), but the same would apply to an anagenetically evolving Daspletosaurus; should D. wilsoni represent a descendant of D. torosus as proposed here, then all of the synapomorphies of the D. wilsoni + more derived tyrannosaurines clade must have been evolved within the ~500 kyr window between D. torosus and D. wilsoni (it should be noted that this would also be true in the case of recent divergence of the D. wilsoni + more derived tyrannosaurine clade via cladogenesis; therefore, low bootstrap scores cannot be taken as evidence of anagenesis, but are at least consistent with it).

More generally, as sampling of a lineage increases, the temporal windows between sampled taxa must necessarily be reduced, and synapomorphies of progressively more derived clades will therefore be fewer (with the same number of character changes distributed among more clades as sampling increases), such that bootstrap scores can be expected to correlate negatively with sampling intensity. To this point, removal of D. wilsoni (in addition to Nanuqsaurus and Thanatotheristes, as described above; see Results) from the cladistic analysis recovers an identical tree topology, but increases bootstrap support for the D. horneri + more derived tyrannosaurines clade from 8 to 25 (still a low score, but over three times higher).

Bootstrap scores can also be affected by the inclusion of characters irrelevant to the node in question (Soltis & Soltis, 2003). The phylogenetic character matrix of Carr et al. (2017) used here contains characters informative across Tyrannosauroidea, including hundreds of characters that are not informative within Daspletosaurus or derived Tyrannosaurinae in general (i.e., characters not recovered as autapomorphies or synapomorphies for species or groups within this clade, respectively). We therefore regard the low bootstrap scores recovered for the phylogenetic placement of D. wilsoni not as evidence of an erroneous result, but as an expected consequence of higher taxonomic resolution among derived tyrannosaurines and the nature of the data matrix used.

Conclusions

D. wilsoni sp. nov., a stratigraphic and morphological intermediate between D. torosus and D. horneri, is hypothesized to represent a transitional form along an anagenetic lineage linking both previously named species of Daspletosaurus. This finding, in concert with previous identifications of anagenesis in contemporary dinosaur lineages, emphasizes the explanatory power of anagenesis in the production of evolutionary trends among dinosaurs of the Late Cretaceous Western Interior (Scannella et al., 2014; Freedman Fowler & Horner, 2015; Fowler & Fowler, 2020; Wilson, Ryan & Evans, 2020). Indeed, as anagenesis continues to be identified among fossil lineages, the predominant relative frequency of strictly cladogenetic evolutionary models (e.g., punctuated equilibria; Eldredge & Gould, 1972) must eventually come under scrutiny. Future explorations of evolutionary mode in fossil taxa, including further tests of the hypotheses presented here, will be important in this regard, and have the potential to refine understanding of the pattern and process of dinosaur evolution.

Supplemental Information

Supplemental Information 1 Character Data and Supplemental Figure.

Click here for additional data file.

Supplemental Information 2 Character matrix for cladistic analysis.

Click here for additional data file.

Special thanks are given to John Wilson for his discovery of the holotype specimen. Thanks also to Elizabeth Freedman Fowler and Matthew Lavin for discussion and guidance that improved the quality of this manuscript, and to David Hone, Jared Voris, and Thomas Holtz for helpful reviews. Thanks to Joshua Chase, Greg Liggett, Pat Gunderson, and other employees of the Bureau of Land Management who assisted DF with land access, permitting, and excavation. We would also like to thank the members of BDM field crews that worked tirelessly to excavate “Jack’s B2,” including Meara Aberle, Jon Carr, Andrew Chappelle, Steven Clawson, Chalfant Conley, Brian Conway, Joshua deOlivera, Jordan Drost, Robert Ebelhar, Elizabeth Flint, Elizabeth Freedman Fowler, Joshua Fry, Sandi Guarino, Brad Hoole, Felipe Jannarone, Marianna Karagiannis, Ashley Lambert, Rachel Livengood, Kat Maguire, Marianna Rogers, Emily Waldman, Solin Wanders, Alyssa Wiegers, Jack Wilson, and many others, without whom study of the holotype specimen would not have been possible. Thanks to Steven Clawson, Amanda Hendrix, Destiny Wolf, and Darrah Steffen for their preparation and curation of the holotype specimen. Thanks also to the Bergtohl family for land access, and to Matt Poole and Montana State Office Glasgow for camping access.

Institutional Abbreviations

AMNH American Museum of Natural History, New York, New York, USA

BDM Badlands Dinosaur Museum, Dickinson Museum Center, Dickinson, North Dakota, USA

CMN Canadian Museum of Nature, Ottawa, Ontario, Canada

FMNH Field Museum of Natural History, Chicago, Illinois, USA

GIN Palaeontological Centre of the Mongolian Academy of Sciences, Ulaanbaatar, Mongolia

GM Ganzhou Museum, Ganzhou, China

MOR Museum of the Rockies, Bozeman, Montana, USA

TMP Royal Tyrrell Museum of Paleontology, Drumheller, Alberta, Canada

UALVP University of Alberta Laboratory for Vertebrate Palaeontology, Edmonton, Alberta, Canada

UMNH Natural History Museum of Utah, Salt Lake City, Utah, USA

ZPAL Institute of Palaeobiology of the Polish Academy of Sciences, Warsaw, Poland

Additional Information and Declarations

Competing Interests

Author Contributions

Data Availability

New Species Registration

The authors declare they have no competing interests.

Elías A. Warshaw conceived and designed the experiments, performed the experiments, analyzed the data, prepared figures and/or tables, authored or reviewed drafts of the article, and approved the final draft.

Denver W. Fowler conceived and designed the experiments, prepared figures and/or tables, authored or reviewed drafts of the article, and approved the final draft.

The following information was supplied regarding data availability:

The holotype specimen (BDM 107) is stored in the collections of the Badlands Dinosaur Museum in Dickinson, North Dakota.

The matrix used in the cladistic analysis is available as a Supplemental File.

The following information was supplied regarding the registration of a newly described species:

Daspletosaurus wilsoni sp. nov. species LSID: urn:lsid:zoobank.org:act:790497D0-3664-4F83-BB13-67CDB27B5428.

Publication LSID: urn:lsid:zoobank.org:pub:F7EE2619-89FC-4D72-93DA-EFE6BD549A77.

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
