# Peer review of "A transitional species of Daspletosaurus Russell, 1970 from the Judith River Formation of eastern Montana"

_PeerJ, doi:10.7717/peerj.14461_

## Round 0.1 · original submission · Major Revisions

Three reviewers have provided a variety of detailed and constructive comments for your manuscript. Most of the comments are in the minor to moderate revision realm, but some (if implemented or considered valid) may require a greater adjustment to the structure of the paper and its conclusions, depending on how you choose to implement them. So, I have selected "major revisions" as the category.

SIGNIFICANT POINTS FROM THE REVIEWERS:

- Two reviewers note that individual voucher specimens, or at least literature citations, should be referenced for all of the anatomical comparisons. Given the changing nature of taxonomic assignments over time, this is an essential point that must be updated in the revision.

- Two reviewers also note that a basic ontogenetic assessment should be added--even just a reference to skeletal fusion would be helpful, of course recognizing the limitations of such methods. I do not view osteohistology as essential for the current paper (although of course it would be great in future work!).

- One reviewer provides information for additional comparison with Daspletosaurus torosus, including additional similar or shared features. This should definitely be addressed in revision (regardless of whether or not you think it is the same taxon as D. diadematus). A second reviewer expresses some thoughts that D. diadematus may be the same as the Dinosaur Park Formation organism (e.g., FMNH PR 308), or that it is at least something for a point of comparison. I do agree that even a little bit of additional comparison is appropriate, but also recognize the need to respect ongoing work.

- Regarding the reviewer comment about the testability of anagenesis, I think it is sufficient to briefly note it (as already done to some extent), provide a minor amount of expansion in response to the reviewer's comments, and leave it at that.

COMMENTS FROM THE EDITOR:

- During revision, I suggest adding some additional information to the abstract -- the submitted version provides a reasonable summary of the paper's conclusions, but should also provide some of the information that *supports* the conclusions, and basic data about the holotype. What is its age? Where does it fit between other Daspletosaurus species? What are unique features, and transitional features? etc. The abstract should be able to stand alone as a capsule summary of the paper itself.

- The photographic figures are very nicely done! They do an excellent job of supporting the text and documenting the morphology of the fossil.

- In Figure 13, I would consider adding range / uncertainty bars on each taxon. The star for D. diadematus implies that the age is *precisely* 76.4 Ma, whereas the text indicates 75.64 Ma as an absolute upper limit, and ~76.5 Ma is the most likely. Is there an absolute lower limit? This does not affect your conclusions, of course--but it would be good to clarify in the figure.

- Consider (optionally) displaying the phylogeny in Figure 14 in time-calibrated format, with ranges/estimated ranges for each taxon. This would help to better depict the hypothesis of anagenesis, too.

- Instead of removing Nanuqsaurus and Thanatotheristes prior to analysis, consider producing a reduced strict consensus tree instead by removing those two post hoc--does it affect reconstructed topology relative to the one where they are removed before analysis? This is especially important given the previously hypothesized close relationships between these two taxa and Daspletosaurus, and if they affect topology within the Daspletosaurus+Tyrannosaurus etc. part of the tree. As noted by one reviewer, the full trees could be provided in summary form (perhaps as supplemental information).

- Optional: give strong consideration to adding additional measurements for major bones -- e.g., the dentary, quadrate, etc. When an element such as the dentary is described as "deep," numerical support should be provided. Deep relative to what? How does this animal compare in overall size to other Daspletosaurus specimens? You can presumably estimate overall skull length, and this might be included (optional).

·

Basic reporting

I have various small issues with the writing and some comments that I think will help improve the MS and things that can be made clearer, especially in the figures. In general this is good, but I do have a couple of bigger points that seriously need addressing.

There are though some more major issues that need attending. These should be relatively easy to correct but absolutely need doing.

1. There's been disagreement over what even Daspletosaurus is in the recent literature and you don't really mention this till you get to the discussion and say you don't consider the Dinosaur Park Formation material to be Daspletosaurs. This needs to come in the introduction and be much more detailed. As written I got through the whole of the definition, diagnoses and comparisons before I found out you have a different concept of the taxon to me. That's extremely unhelpful. It must come earlier and be more detailed.

2. There's essentially no consideration of ontogeny in the paper. At least some of the characters you describe and use vary ontogenetically and this needs to be considered. You also don't say anything about the ontogenetic status of your specimen which is then really problematic in this context. I know it can be hard to assess from incomplete remains but even based on size and things like the fusion of the cervicals and sacrals should give you a ballpark idea of the status of the animal and make for much more meaningful comparisons.

3. Most importantly, you main description is almost completely devoid of sources. You make numerous comparisons to other other taxa (e.g., present in Tyrannosaurus, larger than in Tarbosaurus etc.) with almost no citations of any papers or direct references to specimen numbers (even more oddly, you have a list of Institutional Abbreviations at the end of the MS which never come up because there's no references to specimens). This needs to be addressed - there's simply no traceable sources for pages of your comparisons and it's a massive omission.

Experimental design

I can't comment in detail on this but it comes over as a reasonable extension of the Carr et al. analyses o D. horneri.

Validity of the findings

As per the above comment.

Additional comments

There are places where this can be made more clear. For example the final figure looks nice but it's simply three skulls on a timeline. Listing the relevant features and even showing them on those skulls would suddenly make this an incredibly informative and powerful explanation of the argument being made when right now it's just a couple of drawings. It's not needed but it would massively help to show the case the authors want to make.

·

Basic reporting

To the editor,
Please find included my review of the manuscript by Washaw et al titled “A transitional species of Daspletosaurus Russell, 1970 from the Judith River Formation of eastern Montana”. The manuscript has the potential to augment our understanding of Daspletosaurus stratigraphy, geography, and potentially tyrannosaurid evolution, however, several key issues will need to be addressed prior to the manuscript being publication ready. Most notably, I have some concerns that the newly proposed holotype may actually represent an individual of Daspletosaurus torosus rather than a distinct species. This rationale is based on the specimen being very similar to the few Daspletosaurus torosus specimens that are known with many of the proposed autapomorphies of D. diadematus even being seen in the D. torosus holotype. While I do agree with previous authors (incl. Currie, 2003 and Paulina-Carabajal et al., 2021) that Daspletosaurus specimens from the Dinosaur Park Formation (as well as those from stratigraphically equivalent layers within the Oldman Formation south of Dinosaur Provincial Park) are morphologically distinct from those of Daspletosaurus torosus from stratigraphically lower deposits of the Oldman Formation within Dinosaur Provincial Park, the newly proposed species is more similar to the latter than the former. I have included photographs below that may assist the authors in their research and would be more than happy to provide additional photos or other materials upon request.
It is important to note, however, that the lack of distinctiveness of ‘D. diadematus’ from D. torosus does not detract from the scientific importance of the discovery of the proposed holotype BDM 107. The stratigraphic and geographic provenance of the specimen are both critically important data points in the study of tyrannosaurid biology. If the proposed age ranges are accurate (and there are no glaring reasons to suggest they are not), the stratigraphic provenance of the BDM 107 would suggest that D. torosus was present during the interval of time for which the Dinosaur Park Formation was being deposited. This could suggest a coeval occurrence of Daspletosaurus torosus within Montana and Dinosaur Park Formation Daspletosaurus sp. Additionally, this specimen represents the most southeasterly occurrence of Daspletosaurus torosus and potentially for all Daspletosaurus spp.
For the reasons provided, I propose the manuscript to be in need of major revisions. There are some key issues that need to be addressed, however, the study itself is still scientifically significant and therefore is still worthy of publication pending the addressing of these issues. I have included an edited version of the manuscript document which details more specific issues.

Experimental design

no comment

Validity of the findings

see above

Additional comments

Additional annotated documents are included

Additional Comments
- You’ll need to rerun the phylogenetic analysis after addressing the relevant concerns regarding characters and scorings. Be aware, this may alter possible evolutionary scenarios and may therefore need to be addressed in the discussion as well.
- Much of the description is under cited. The authors make several comparisons with other taxa that are not the focus of this study using features that have been previously discussed in the literature. Please add relevant citations to observations/features that are not your own.
- Many parts of the description could benefit from more detail. You have quite a bit of material but tend to be rather brief in your descriptions. For example, you have a very well-preserved postorbital but focus almost entirely the cornual process all while there are plenty of other useful features in this bone including within the ventral ramus and frontal process.
- Grammar and structure could use some work. E.g., many paragraphs are quite short (often being comprised of a single sentence). Try to avoid this where possible. I’ve marked places where most relevant in the text.
Figure Notes
- Try increasing the brightness on figure 10. As is, its somewhat difficult to see the features you discuss. Also maybe try using a more angled light source to better show off the topography of the bone.
- Some figures could benefit from having additional angles (particularly of structures discussed in the text). For example, an anterior view of the postorbital and ventral view of the squamosal.
- Figure 12 caption states the graphic represents a time calibrated phylogenetic tree however it is not a phylogeny. Please fix.

·

Basic reporting

The authors describe new material of the tyrannosaurine dinosaur Daspletosaurus and provide evidence that this specimen, stratigraphically intermediate between the previous named species, represents a new transitional taxon in this sequence. They further posit that the character evidence along with the stratigraphic data supports an anagenetic model of evolution of Campanian tyrannosaurinins (and potentially onward into Maastrichtian forms).

The description of the fossil material itself is good, and the fact that they highlight the non-plesiomorphic information in each element is very helpful. The illustrations compliment these descriptions. One thing that is missing is an assessment of the ontogenetic status of the individual: in several of the papers cited in this manuscript previous authors have made it abundantly clear that there can be substantial transformation of character traits during ontogeny within Tyrannosauridae. Thus, being able to compare this individual with specimens of the same ontogenetic status in other species would be very useful.

Experimental design

I am mildly concerned that if this is a new species that it might be the same as the Dinosaur Park Formation species which Currie and colleagues have proposed over the past several decades but have yet to formally describe. (The best example of this alleged species is FMNH PR 308, which would presumably become the holotype.) This species has been included as an operational taxonomic unit in at least one of the studies cited here (Loewen et al. 2013, as the “Dinosaur Park Formation tyrannosaurid”.) On the one hand, it is up to Currie et al. to get off their collective butts and finally describe it and justify it if it is a new species, and if D. diadematus preempts their work, it is their own damn fault for waiting. That said, given the likely temporal overlap between it and Daspletosaurus diadematus a comparison of the two specimens is very much justified.

Line 322 discusses the analysis when fragmentary taxa were included. If possible, please provide this tree. It is fine to provide a messy-but-more-inclusive tree as well as the more legible and better resolved one for comparison. Also, please state which mode of consensus was used: strict, Adams, and majority-rule consensus trees can provide radically different topologies (I won’t say “results”, as consensus trees are not results: they are meta-result summaries of the actual results.) If memory serves, TNT’s default consensus trees are strict consensus. Additionally, please make sure to state you were using TNT for the search and cite the software in the references.

Validity of the findings

The authors present morphological traits that are within previous practice of naming new dinosaur species. Of course, that is entirely a different statement than saying these species are actually justified in a biologically meaningful context. Carr et al. (2022) show that you can have considerable variation in a set of individuals in a dinosaur taxon but lack the statistical justification in subdividing them into multiple species, and that was done with a much larger sample size than given here. So it is entirely possible that future studies will lump one or more Daspletosaurus species into a single one; that is up for future studies to show.

The authors make bold statements about this new data as rejecting cladogenesis and punctuated equilibrium, but in fact their data is too sparse to show that. The hypothesis presented (that there are a series of “Daspletosaurus” species which form an anagenetic series with respect to Zhuchengtyrannus, Tarbosaurus, and Tyrannosaurus) is not unreasonable, and a version of this hypothesis was presented in Honer et al. 1992 and Loewen et al. 2013. However, even if the phylogeny is accurate, it is also consistent with a branching sequence. Given that 8 of the 11 recognized non-alioraminin tyrannosaurine species (Teratophoneus curriei, Dynamoterror dynastes, Lythronax argestes, Daspletosaurus horneri, Nanuqsaurus hoglundi, Thanatotheristes degrootorum, Zhuchengtyrannus magnus, and the new Daspleteosaurus diadematus) were named only in the 2010s or 2020s we cannot fairly say we actually have a representative complete understanding of the diversity of this clade even in Campano-Maastrichtian Laramidia. Our coverage is simply too sparse at present to saw we have sufficiently covered the relevant species (much less the contemporary Asian ones). The authors do acknowledge some of these limitations in the Discussion, to be certain.

More significantly than the issue of “unknown unknowns”, however, is sample size within the species. The distinction between anagenesis and cladogenesis, and between punctuated equilibrium and phyletic gradualism, is one of both stratigraphic range data and statistical survey of morphology within these ranges. But most of the species involved are from single time slices, with sample sizes of less than one complete individual (much less a statistically significant population-level sample at multiple time slices throughout the species or chronospecies duration.) One of the most recent examples of a fair test concerning an anagenetic vs. cladogenetic model for terrestrial vertebrates was O’Leary 2021, where she measured over 3000 samples from dozens of stratigraphic horizons over a span of a mere 3 million years and was able to tentatively support a pattern of gradual and continuous anagenetic change in notharctine primates. That analysis and its dense data concentration makes her conclusions convincing and well-supported.

Those of us in the dinosaur paleobiology community might dearly desire our chosen taxon to be relevant for this kind of macroevolutionary studies, but we have to recognize the limitation of our dataset. We might be spectacular for addressing biomechanics and other functional morphology, phylogeny and evolutionary transitions, display and behavior studies, and plenty more. (Indeed, we probably have a data set far better for these than those of, for instance, mammalian taxa known only from sets of molars.) But for studies which are ultimately based on extremely large sample sizes extended along stratigraphic ranges, our data is sadly and frankly sucktacular.

(Again: I am not saying the conclusions are wrong; only that we can’t show that they are correct in a meaningful say. They can be a speculation, but the Eldredge/Gould/Stanley/Vrba/etc. model is almost certainly not going to be overturned by Laramidian dinosaur studies, and these are the best we dinosaur workers have to offer. It will have to be with appropriate study taxa, such as foramiferans, mollusks, trilobites, pollen, micromammals, etc. And again, the authors do acknowledge some of the limitations in their Discussion.)

O’Leary, M.A. 2021. A dense sample of fossil primates (Adapiformes, Notharctidae, Notharcinae) from the Early Eocene Willwood Formation, Wyoming: documentation oof gradual change in tooth area and shape through time. American Journal of Biological Anthropology 174: 728-743. Doi: 10.1002/ajpa.24177

Additional comments

This is a helpful contribution to our knowledge of tyrannosaurid paleontology and to Campanian dinosaurian diversity.

---

## Round 0.2 · Minor Revisions

Thank you for your close attention to the comments from the reviewers. Given the significant changes from the first submission, I sent the manuscript out for a second round of review, and two of the original three reviewers were able to provide feedback.

For the diagnosis, is it worth adding in the morphology of the mylohyoid foramen as distinguishing D. wilsoni from other Daspletosaurus? Although one reviewer has some concerns with the diagnosis, I can see this as a matter of taxonomic opinion and/or making minor modifications, and I am quite fine with the species standing (other than making any final modifications to the diagnosis).

There are a few other comments from the reviewers of fairly limited scope; once those are addressed, I should be able to make a decision in very short order.

·

Basic reporting

This is much improved from the previous version though there are numerous small quibbles with the writing and presentation (see marked up document). There's various issues with the references, inconsistencies (cervicals aren't listed as part of the holotype and then later described, you say there are no unique traits in the taxonomy section and then later describe an autapomorphy) and these need fixing.

Experimental design

N/A

Validity of the findings

My real concern here is the taxonomy section. In the initial submission, a total of 5 autapomorphies were listed for the putative new taxon, nbut in this new version that has been revised to be only a combination of features that make it intermediate from the other two. That's not inherently problematic as there is nothing wrong with diagnosing taxa from combined traits, but of the four traits listed that unite it with D. torosus, potential issues are listed with three of them. In short, there is very little here to unite it with torosus and a bunch of traits to unite it with D. horneri.

That makes the idea that this is some clear intermediate with traits shared with both rather questionable. You have 1 solid trait shared with torosus and one autapomorphy and lots of traits shared with horneri. Does that not make this simply a geologicallyyounger specimen of D. horneri with a couple of odd traits (no big surprise when you have most of a well preserved skull and some traits will inevitably vary if you look at enough). In short, I'm not convinced *based on the characters as now presented* that this is a) new or b) really a convincing intermediate and that really affects the paper as a whole. If you can shore this up that will solve the problem but if not, then much of the interpretation becomes really problematic.

Additional comments

N/A

·

Basic reporting

This revision is greatly improved over the earlier incarnation of the manuscript. The majority of the concerns of myself and the other two reviewers seem to have been adequately dealt with.

Experimental design

The methods discussed are consistent with standard practices.

Validity of the findings

The authors have greatly improved their case with the new data and more detailed description of the fossil material and of their analyses.

Additional comments

Line 620: Two issues with the use of "500 Ka". Firstly, the SI prefix for 10^3 is a lowercase "k", not an uppercase "K": hence, kg, km, kbar. Also, the community standard in geology is to use "ka", "Ma", "Ga" for dates before present, and "kyr", "Myr", "Gyr" for durations in the past. A relatively recent paper which discusses this is Aubry, M.-P. et al. 2009. Terminology of geological time: establishment of a community standard. Stratigraphy 6: 100-105 https://citeseerx.ist.psu.edu/viewdoc/download?doi=10.1.1.495.8644&rep=rep1&type=pdf

As per one of the other reviewers, I do not think "phylogeny" is the best term for the figure portrayed. "Sequence of chronospecies" might more accurately reflect the context of the graph.

Finally, with regards to the discussion running from 553 to 641: once again I make my plea that the authors not consider a study involve a few sparse individuals spread over a substantial chunk of geologic time as a fair test of phyletic gradualism vs. punctuated equilibrium. For instance, lack of polytomies can be consistent with anagensis, but it is also equally consistent of sparse data and lack of sufficient samples to provide character conflict or ambiguity. The strength of the polytomy argument comes in when a sufficiently large representation of all of the species involved are known: a situation unlikely for Judithian dinosaurs (and indeed the authors admit they are excluding at least one relevant taxon in this: the Dinosaur Park Formation taxon.)

Additionally, phyletic gradualism vs. punctuated equilibrium is much more than anagensis vs. cladogenesis. The latter is more specifically the argument that the majority of morphological change between species occurs in relatively short bursts (finally defined by Gould 2002 as "10% or less of the species duration" [Gould, S.J. 2002 The Structure of Evolutionary Theory. Bellknap Press.]) and lacking significant directional change over most of the species duration. In contrast, phyletic gradualism allows for directional change throughout the entire species duration and no particular significant episodes of morphological shift.

The Daspletosaurus data is thus wholly inadequate to address those issues, as the samples of each are just a few isolated individuals. However, there is no need to make this important new fossil or even the Laramidian dinosaur data be the test of grand evolutionary theories.

It is much fairer and fully supported by the data shown here that the observations and phylogenetic pattern is consistent with anagensis. And furthermore the data (at least at present) lack one of the predictions of cladogenesis: the persistence of the ancestral species above the first appearance datum of the descendant. Such a situation is nonsensical in a case of anagensis, as the population as a whole has shifted and the "speciations" are arbitrary cuts of a gradually changing succession. But it is a possible situation where one species buds from another. As the authors here have shown, the Judithian tyrannosaurine data is entirely consistent at present with an anagenetic sequence of chronospecies.

---

## Round 0.3 · accepted · Accept

Thank you for your thorough attention to the most recent round of reviews. In my view, the manuscript is now ready to move forward.